# Estimating the gains of early detection of hypertension over the marginal patient

**Paul Rodríguez-Lesmes** *

School of Economics, Universidad del Rosario, Bogotá, Colombia

* paul.rodriguez@urosario.edu.co

## Abstract

This study estimated the potential impact of early diagnosis programs on health outcomes in England. Specifically, if advising individuals to visit their family doctor due to a suspected case of mild hypertension would result in (i) an increase in the diagnosis and treatment of high blood pressure; (ii) an improved lifestyle reflected in objective measures such as the body-mass-index and blood pressure levels; (iii) a reduced probability of the onset of other cardiovascular diseases, such as diabetes. To address potential selection bias in screening, a feature of the English Longitudinal Study of Ageing is exploited, motivating a regression discontinuity design. If respondents' blood pressure measurements are above a standard clinical threshold, they are advised to visit their family doctor to confirm hypertension. Two years after the protocol, there is evidence of an increase in diagnosis (5.7 pp, p-val = 0.06) and medication use (6 pp, p-val = 0.007) for treating the condition. However, four years after the protocol, the difference in diagnosis and medication disappeared (4 pp, p-val = 0.384; 3.4 pp, p-val = 0.261). Moreover, there are no differences on observed blood pressure levels (systolic 0.026 mmHg, p-val = 0.815; diastolic -0.336 mmHg, p-val = 0.765), or Body-Mass-Index ((0.771, p-val = 0.154)). There are also no differences on diagnosis of diabetes (1.7 pp, p-val = 0.343) or heart related conditions (3.6 pp, p-value = 0.161). In conclusion, the nudge produces an earlier diagnosis of around two years, but there are no perceivable gains in health outcomes after four years.

## 1 Introduction

Globally, preventive care has become the main instrument of public health policy. The reason for this is the surge in non-communicable diseases (NCDs) as the main cause of death around the world. Almost 70% of all global deaths are due to NCDs: a third of those were individuals aged 69 years or younger [1]. This pandemic is explained mainly by lifestyle factors: smoking, alcohol consumption, and lack of physical activity and exercise. Several of those diseases start with latent conditions, such as high blood pressure (HBP); thus, to address the problem, the implementation of regular medical checks has been proposed as an appropriate public health policy. Lack of information about underlying health status is cited as a reason for puzzling empirical facts in light of Grossman's health capital model [2]. In essence, if individuals obtain accurate information about their health, they adjust their lifestyles accordingly.

**Data Availability Statement:** This article uses publicly available data from the English Longitudinal Study of Aging, administered by NatCen (https://www.elsa-project.ac.uk/accessing-elsa-data). All codes needed for replication of the

exercises are available on GitHub (https://github.com/androdri1/earlydiagnosis).

**Funding:** The author received no specific funding for this work.

**Competing interests:** The author has declared that no competing interests exist.

Notwithstanding the high expectations, results on screening programs for such conditions as hypertension have been mixed [3]. That has been the case with large-scale programs, such as *Health Checks* of the UK National Health Service: that had an annual cost of around 450 million GBP, but the impact was modest [4–7].

This paper studies the causal link between lifestyle and the acquisition of health information in England. In particular, the paper estimates the impact of informing an individual about their likely hypertensive status while dealing with the empirical challenge of screening selection. This challenge may arise through at least two reasons. First, individuals who value their health more also demand more preventive care services and maintain healthier lifestyles. Second, individuals who expect their future health to deteriorate faster that the average person may adopt a different lifestyle and have greater demand for preventive health services. To deal with that challenge, a regression discontinuity design (RDD) is implemented based on differential feedback given according to cutoffs in blood pressure (BP) measured as part of a longitudinal survey protocol. This study considers a sample that was aged on average 53 years; who were visited again when their mean age was 55 (range, 50–60) years, and who had a final review when the mean age was 57 years.

Within the testing protocol of the *English Longitudinal Study of Ageing* (ELSA) [8], during the collection of BP measurements by a nurse, respondents with a record above a defined BP threshold were advised to visit their family doctor: they could have been suffering from HBP. As a result, it was possible to compare lifestyles, health perceptions, medication use, and biomarkers of individuals just below and above the threshold 2 and 4 years after the initial measurement. With that strategy, it is possible to obtain causal estimates for individuals not previously diagnosed with, or under pharmacological treatment for, HBP or diabetes and who were very similar in their health status and lifestyles but who differed only in having being advised, or not, to visit their family doctor owing to suspected hypertension.

Given the design, the study tests the following three hypotheses:

1. The advice would result in an increase in the diagnosis of HBP and on the treatment for it (ex. medication usage).

2. The advice would result in a better lifestyle, reflected on lower levels of BMI (showing a change either on caloric intake or on exercise levels) and blood pressure.

3. The advice would reduce the probability of the onset of other cardiovascular diseases, such as diabetes.

This paper contributes to two main strands of the health information shocks literature.

First, it contributes to the growing literature on marginally ill patients. Essentially, the analysis of characteristics of patients which receive information on a particular condition, but whose biomarkers indicate a borderline result with respect to the standardised cutoffs. For instance, Almond et al found that telling secondary students that they are over-weighted, following the BMI-based definition, does not help to reduce the weight of slightly over-weighted students [9]. Also, Dahlberg et al. studied the case of revealing information about abdominal aortic aneurysm, where cutoffs are also used to determine the level of surveillance needed [10]. The authors found that well-being is not affected by the potential illness notification, but it is when respondents are classified to the in a risky position. With respect to health checkups, there are two studies in this line that are our main references. First, Zhao and coauthors [11]. They present a similar identification strategy, a RDD based on blood pressure diagnosis thresholds, which is used on a Chinese panel to assess the impact of information on health along income distribution. They found a reduction on fat intake for those who were informed about having hypertension. Second, Iizuka et al considered a similar design using a mandatory

screening program for the case of diabetes in Japan [12]. They found that medical utilisation increased but no objective benefits were found by the diagnosis of borderline diabetes diagnosis. Similar analysis, and results, for diabetes where considered for the US and Korea [13, 14]. This research complements this literature by considering the case of advise (rather than diagnosis) of HBP in England, where the prevalence of hypertension is high and where early diagnosis programs have been implemented as public policy. Also, given the higher frequency of ELSA waves, it was possible to assess both the short- and long-term impacts of medical advice.

Second, this paper adds on discussions about periodical screening programmes and their potential for inducing changes on demand for health care. These programs nudge individuals into formal screening procedures [15]. The findings our this research are in line with previous findings in the literature on health checks, which found an increase in medication use [3, 16]. Robson and coauthors suggest that NHS Health Checks is related to an increase in the attendance to general practitioners (GP) practices of individuals in risk of developing CVDs [5]. They observed an increased prescription of medication for controlling high blood pressure and lowering cholesterol. Authors such as MacAuley criticise periodical screening programmes not only for their potential misallocation of resources and over-diagnosis of certain conditions, but also because they offset individual efforts in terms of their health-related behaviour [17]. This paper shows that such impacts are not permanent, and as a result their effectiveness have to be consider in terms of time windows.

The rest of this paper is organised as follows. Subsection 2.1 presents the particularities of HBP diagnosis, prevalence, and treatment in the United Kingdom; it also describes the theoretical framework through which it is possible to understand the role of information shocks in a competing risks model with multiple health investments. Subsection 2.2 addresses the main details of the dataset and explains the health advice procedure implemented by the survey nurses. Subsection 2.3 discusses the empirical strategy. In subsection 2.4, the study sample characteristics are presented as well as the validation of the RDD strategy. Section 3 covers the results, and section 4 presents the discussion and conclusions.

## 2 Material and methods

### 2.1 High blood pressure in the UK

Worldwide, controlling BP levels is one of the most cost-effective methods for reducing premature mortality [18, 19]. Elevated BP results in progressive vascular and renal damage, which increases the risk of cardiovascular diseases (CVDs), such as ischemic and hemorrhagic stroke and myocardial infarction [20]. Such is the importance for preventive care, that some countries, such as the United Kingdom, provide financial incentives to primary care physicians for improving BP management. In the UK, between 1994 and 2011, around 30% of adults aged 30 years and older had HBP [21]. Despite important improvements over time, 13% of adults still had untreated hypertension in 2011. In 2016, that proportion was almost the same [22].

In the United Kingdom, diagnosis is based on the cutoff of 140/90 mmHg systolic / diastolic BP (SBP/DBP). The severity is classified in three stages: stage 1 begins at 140/90 mmHg; stage 2 is at 160/100 mmHg; and the severe stage 3 is where SBP is 180 mmHg or above, which requires an immediate intervention [20]. However, BP notoriously varies during the day and in different situations. In particular, there is the well-known white-coat effect: records can be even 20/10 mmHg higher when measured at a clinic [23]. According to guidelines, if an individual has a reading above any of the cutoffs at the clinic, ambulatory blood pressure monitoring is used to confirm the result: the patient wears a portable monitoring device for at least 4 days, and the average of at least 14 measurements is compared with the cutoffs [20]. For the general population, clinic-based measures are 4/3 mmHg higher than ambulatory measures

near the stage 1 HBP thresholds [24, 25]. In some scenarios, even if differences in prevalence are not conspicuous, non-ambulatory measures have lower sensitivity and specificity [26].

In terms of treatment, to reduce cardiovascular risk, guidelines place special emphasis on the importance of advice on lifestyle modifications [20, 27]: smoking cessation, improved diet, weight loss, reduced alcohol consumption, and increased physical activity. In fact, current guidelines suggest offering BP-lowering medication to stage 1 patients only if other diseases are present or if the 10-year CVD risk (an index based on other characteristics in addition to BP levels) is above 20%. For stage 2, the guidelines suggest antihypertensive drug treatment [20].

The present study aims to determine the effects of receiving advice about potential undiagnosed hypertension; thus, the subject population has to be those at risk of such a condition and less likely to be tested for it. Both SBP and DBP increase with age until 60 years, when DBP starts to systematically decrease [21]. By 2011 in the United Kingdom, the prevalence of hypertension was 28% for the 40–49 age-group, 40% for 50–59, and 60% for 60–69. Detected prevalence was related to the proportion of individuals who regularly underwent BP measurement; that proportion increased over the study period. Between 1998 and 2008, data from the *British Household Panel Survey* indicated an increase of 61%-80% in the proportion of individuals aged 45–60 who reported having had their BP tested in the previous 2 years [28]. The proportion was larger for the older group: 73%-86%. As a result, despite improvements over time, the prevalence is higher in older individuals; however, testing is lower in the middle-age group.

## 2.2 Health advice intervention

The present study uses ELSA data for the years 2002–19 (waves 1 -9) [29]. ELSA is a longitudinal study with a representative sample of individuals aged ≥50 years in England. Its baseline (wave 0) was constructed using the Health Survey for England (HSE) [30], and it contains high-quality subjective and objective health information and socioeconomic details. The characteristics of ELSA make it ideal for this analytic purposes. Nurses hired for ELSA visit survey respondents 2–4 weeks after the survey interview and take BP readings in waves 0, 2, 4, 6, 8 and 9. Further details about the dataset, consent, and access to the data can be found at https://www.elsa-project.ac.uk/data-and-documentation.

According to the ELSA protocol, nurses advise respondents to visit their family doctor (general practitioner [GP]) if at least one of the BP measurements is above a certain threshold. This message may cause some individuals to visit their GP and undergo more comprehensive screening to confirm whether or not they are hypertensive.

DBP and SBP measurements were derived by using the last two of three measurements, in order to minimise the likelihood of the *white coat syndrome*, using an automated monitor under standardised conditions. In ELSA, the thresholds are 140/85 mmHg. In the early HSE (1998, 1999, 2000 measurements), the values were the same for women and men <50 years old; however, the values changed for men aged ≥ 50 years when 160/95 mmHg was used. If a reading was above those thresholds, respondents were instructed to visit their GP within 3 months. Those thresholds are similar to the official recommendation for systolic BP used by Britain's National Health Service (NHS) [31]. As in similar studies for HBP, only the SBP cutoff is considered as physicians usually pay closer attention to that reading [11].

Nurses were clearly instructed to provide only interpretations according to the survey protocol. Respondents were allowed to avoid receiving feedback following the measurements or for the results to be sent to their GP. Unfortunately, there was no information about whether the respondents visited their GP in the weeks following the nurse visit. The recommendation

could be made as a note in the measurement record card along with notes about other bio-markers. There were no other comments or suggestions based on the survey's biomarkers. As stated in the survey protocol, the suggestion made by the nurses was standardised. They would inform the respondent as follows:

> *Your blood pressure is a bit high today. Blood pressure can vary from day to day and throughout the day so that one high reading does not necessarily mean that you suffer from high blood pressure. You are advised to visit your GP within 3 months to have a further blood pressure reading to see whether this is a once-off finding or not.*

## 2.3 Empirical strategy

The previously described nurse protocol motivates a RDD. The idea therein is to compare the value of the outcomes in the post-measurement waves between individuals just below and just above the threshold. In this way, it is assumed that having the SBP measurement slightly above or below the advice cutoff on the day of the nurse visit is essentially random (a more detailed discussion in presented below). The advantage with this strategy is that the diagnosis of hypertension status is often endogenous in a non-experimental setup [11]. The message would be new information only for individuals previously unaware of their elevated BP; thus, the sample was restricted to respondents who did not report either having been diagnosed with HBP by a doctor or having taken medication to lower BP. Further, individuals diagnosed with diabetes whose BP was normally controlled were excluded. Finally, early detection programs target middle-age individuals since they are more likely to benefit from the detection of asymptomatic conditions. For that reason, the sample was restricted to individuals aged 58 or younger at the time of measurement. In Appendix A in S1 Appendix, the sensibility of the analysis to these restrictions is studied.

Fig 1 summarises the strategy followed in this study. At any given even wave the SBP was contrasted against the cutoffs for respondents who met the inclusion criteria outlined above.

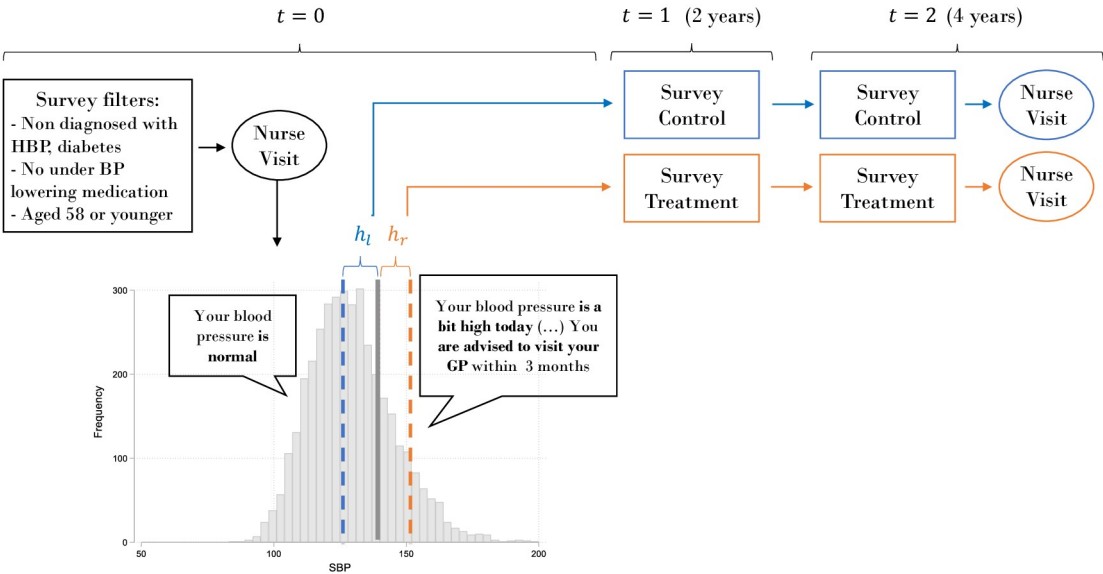

**Fig 1. Systolic blood pressure distribution and nurses' advice.**

HSE has other thresholds, so SBP was centred around the relevant threshold, *sBP*. Individuals just below the relevant cutoff were considered the *control* group ($sBP < 0$) and those just above the *treatment* group ($sBP \geq 0$). Then, the average outcomes (health and lifestyle variables) for both groups in one wave (1–2 years) and two waves (3–4 years) ahead were contrasted. Given the structure of the dataset, biomarker information was available again only two waves later.

Formally, the impact of a nurse's advice at around 2 years ($t = 1$) and 4 years ($t = 2$) of the intervention, $W = 1(sBP \geq 0)$, on outcome $Y_t$ was identified by the discontinuity in the conditional expectation of such outcome at the advice cutoff $sBP = 0$ [32]. The main results are presented based on the estimated parameter $\delta_t$ in the following equation,

$$Y_{i,t} = \delta_t W_i + \alpha_0 + f_l(\alpha_l, sBP_i | W_i = 0) + f_r(\alpha_r, sBP_i | W_i = 1)$$
$$f_x(\alpha_x, sBP_i | W_i = 0) = \alpha_{x,1} sBP_i + \alpha_{x,2} \qquad , sBP_i \in [-h_l, h_r] \tag{1}$$

where functions $f(\cdot)$ represent local linear regressions with triangular weights. Alternative specifications are presented in B. As stated above, if the RDD strategy is valid, in absence of the advice, the average outcome of respondents both *slightly above and below* the threshold would be the same -or at least follow a stable trend. For that reason, it is crucial to define precisely what it is meant by 'slightly above and below'. That is the role of the bandwidths $h_l$ and $h_r$, which were selected based on minimising the mean square error of the non-linear approximation developed by Calonico et al [33]. Hence, the size is a function of the available sample size and local distribution of the dependent variable. That was done separately for the left and right sides because the sample size differed on either side (see the histogram in Fig 1). Also, confidence intervals which are robust to the sensitivity of the bandwidth selection are considered [33]. The RDD estimator was corrected for the potential large selection of a bandwidth and then constructed using a bias-corrected t-statistic. This procedure was implemented using the `rdrobust` routine in STATA [34].

Here, the impact of an alert about potential HBP on health and lifestyle outcomes was identified. This was equivalent to an open screening program, such as NHS checks for individuals with BP levels close to stage 1 hypertension thresholds. That was an important proportion of the reference population (see the histogram in Fig 1); it is particularly important for a health system since it is a key risk factor of CVDs that can be easily managed if detected in time. It should be noted that the estimated impact here may differ from that of a program directed at the general population; the latter type of program would cover normotensive individuals, people with stage 2 or higher hypertension, and those will very low BP levels. The estimated impact also differs from the effect of diagnosis of hypertension.

In the ELSA setup, it is clear that respondents are warned only about the *possibility* of having HBP; it is acknowledged that the BP for diagnosis (based on monitoring BP for at least 4 days) may differ from BP in the survey measurement. Thus, the effect estimated here is a mix between the effect on individuals who received an HBP diagnosis, those who recorded relatively high BP levels but not sufficient to be declared hypertensive, and those who did not follow the advice to visit their family doctor. Unfortunately, the effect among these three groups cannot be identified since ELSA does not record whether individuals visit their doctor after the nurse visit.

## 2.4 Validity of the RDD and sample characteristics

As summarised in Fig 1, the sample selection procedure for this exercise was determined by respondents aged 58 or younger that (1) had no prior history of HBP or diabetes, (2) were surveyed in the wave following the nurse visit, and (3) for whom there were valid SBP

**Table 1. Respondents' characteristics at study entry wave.**

| Variables | (1) | (2) | (3) | (4) | (5) | (6) | (7) | (8) | (9) |
|---|---|---|---|---|---|---|---|---|---|
| | ALL | Below cutoff, bandwidth of | | | Above cutoff, bandwidth of | | | Balance test | |
| | | [-22.5,0) | [-15,0) | [-7.5,0) | [0,7.5) | [0, 15) | [0,22.5) | Difference | [p-val] |
| **Panel A. Socio-demographic** | | | | | | | | | |
| Age | 52.67 | 52.67 | 52.69 | 52.85 | 52.80 | 52.91 | 52.98 | -0.057 | [0.999] |
| Male | 0.39 | 0.41 | 0.40 | 0.41 | 0.39 | 0.38 | 0.38 | -0.033 | [0.393] |
| Educ: Some high level or above qualif. | 0.34 | 0.35 | 0.34 | 0.33 | 0.32 | 0.30 | 0.30 | -0.036 | [0.331] |
| Non white ethnicity | 0.23 | 0.25 | 0.25 | 0.26 | 0.29 | 0.29 | 0.29 | 0.008 | [0.902] |
| Married | 0.75 | 0.76 | 0.76 | 0.75 | 0.76 | 0.75 | 0.76 | 0.037 | [0.291] |
| **Panel B. Health** | | | | | | | | | |
| BMI: Body Mass Index (kg/m2) | 27.14 | 27.24 | 27.56 | 27.71 | 28.15 | 28.29 | 28.48 | 0.120 | [0.618] |
| Valid mean systolic blood pressure | 128.26 | 127.64 | 131.10 | 134.37 | 141.18 | 143.97 | 145.86 | -0.335 | [0.518] |
| Valid mean diastolic blood pressure | 75.28 | 75.30 | 77.06 | 78.74 | 81.03 | 82.45 | 83.27 | -1.193 | [0.115] |
| **Panel C. Economic** | | | | | | | | | |
| Total yearly income† | 36.87 | 37.12 | 36.89 | 36.66 | 35.82 | 34.79 | 34.63 | 1.391 | [0.615] |
| Total net non-pension wealth† | 420.00 | 414.27 | 403.19 | 381.59 | 436.98 | 442.21 | 422.43 | 173.727 | [0.047] |
| Working | 0.79 | 0.80 | 0.81 | 0.81 | 0.81 | 0.80 | 0.80 | 0.010 | [0.754] |

Notes: Simple averages using HSE 1998,99,00 for wave 0 and ELSA waves 2, 4 and 6. It includes only those aged 58 or younger at the time of the measurement who were not diagnosed with HBP or diabetes, and not taking BP-lowering medication. †Variables measured in thousands of GBP of May 2005 for the benefit unit (family). **ALL**: all respondents at waves 0, 2, 4 and 6 (waves where nurse measurements took place), with the same age and medical restrictions but without considering the availability of valid SBP measurements. For columns 2 to 7, the sample was restricted according to the average SBP of the last two out of three valid measurements. SBP is measured in mmHg and centred around HBP stage 1 diagnosis cutoff: 140 mmHg in general, with the exception of males aged 50 and older at wave 0 (HSE 98,99,00) where it is 160 mmHg. Column 8 estimates the jump in the running variable at $SBP = 0$ using a sharp RDD, and column (9) presents the robust p-value for the null of the jump being zero.

measurements slightly above and below the threshold. As a result, a particular respondent could be recruited into the sample at waves 0, 2, 4, and 6.

Table 1 presents average characteristics of the sample at the time of the BP measurement that resulted from the original advice. An examination was made here as to whether there were substantial differences in sociodemographic and economic characteristics; other health measurements between observations above and below the cutoffs were also assessed for different bandwidths *before* the BP measurements and how they compared with the general ELSA sample. The sub-samples in columns 2–7 of that table are based on the value of optimal bandwidth for HBP diagnosis, which is around 15 mmHg. As a reference, one standard deviation of SBP is close to 20 mmHg, which is the difference between stage 1 and stage 2 HBP. A formal balance test is presented in columns 8 and 9 of the table; there, the main specification in Eq 1 is implemented for those variables. For the sake of brevity in the table, optimal bandwidths, sample sizes, and standard errors are not presented. In general, the variables in panels A, B, and C of the table should be orthogonal to the SBP cutoffs. No particular differences were evident: the data indicated that younger, more educated individuals with higher income generally had lower SBP. Further, SBP was positively correlated with DBP and BMI, as expected. It should be noted that apart from those cardiovascular risk indicators, the sample was not particularly different from the general population in terms of other general characteristics.

Table 2 presents the progression of the dependent variable averages through three waves from inclusion in the study sample. There, only the 15-mmHg bandwidth (columns 3 and 6 of Table 1) is presented for brevity. In the Results section, formal balance tests per variable are presented together with the differences 2 and 4 years after the relevant nurse visit.

**Table 2. Dependent variables' means by stage.**

| Variables | (1) | (2) | (3) | (4) | (5) | (6) | (7) |
|---|---|---|---|---|---|---|---|
| | | t = 0 (Before BP test) | | t = 1 (2 years) | | t = 2 (4 years) | |
| | ALL | [-15,0) | [0, 15] | [-15,0) | [0, 15] | [-15,0) | [0, 15] |
| **Panel A. Sample characteristics** | | | | | | | |
| Age | 52.67 | 52.69 | 52.91 | 55.43 | 55.65 | 57.75 | 57.98 |
| Missing this wave | | | | 0.15 | 0.18 | 0.26 | 0.28 |
| Already death by this wave | | | | 0.01 | 0.01 | 0.01 | 0.01 |
| **Panel B. High Blood Pressure** | | | | | | | |
| Diagnosed HBP ever | | | | 0.06 | 0.15 | 0.13 | 0.28 |
| Takes BP medication | | | | 0.02 | 0.05 | 0.05 | 0.12 |
| **Panel C. Other health outcomes** | | | | | | | |
| Diagnosed Diabetes ever | | | | 0.01 | 0.02 | 0.03 | 0.04 |
| Diagnosed Heart Condition | 0.02 | 0.01 | 0.01 | 0.02 | 0.03 | 0.03 | 0.05 |
| Self-reported GOOD health | 0.82 | 0.83 | 0.82 | 0.84 | 0.81 | 0.85 | 0.83 |
| BMI: Body Mass Index (kg/m2) | 27.14 | 27.56 | 28.29 | 27.02 | 28.80 | 28.03 | 28.81 |
| Valid mean systolic blood pressure | 128.26 | 131.10 | 143.97 | | | 129.74 | 138.70 |
| Valid mean diastolic blood pressure | 75.28 | 77.06 | 82.45 | | | 77.09 | 80.95 |
| Number of observations | 305111 | 9048 | 2041 | 1030 | 2041 | 1030 | 2041 |

Notes: Simple averages. It includes only those aged 58 or younger at the time of the measurement who were not diagnosed with HBP or diabetes, and not taking BP-lowering medication. **ALL**: all respondents at waves 0, 2, 4, 6, 8 (waves where nurse measurements took place), with the same age and medical restrictions but without considering the availability of valid SBP measurements. The sample was restricted according to the average SBP of the last two out of three valid measurements at $t = 0$ SBP is measured in mmHg and centred around HBP stage 1 diagnosis cutoff: 140 mmHg in general, with the exception of males aged 50 and older at wave 0 (HSE 98,99,00) where it is 160 mmHg.

Additionally, to assess the selectivity of the nurse-visited sample, column 1 in Table 2 presents the average for the relevant population without considering the availability of SBP information.

First, Panel A in Table 2 presents three key characteristics of the sample. The first is age, which shows the roughly 2-year difference between measurements. The second is attrition, which after 4 years was almost 30%: it was somewhat higher for individuals on the right-hand side of the cutoff. That is a common factor in retirement studies: there is a negative gradient with attrition with respect to education and income -variables that are positively correlated with SBP, as evident in Table 1 [35]. Thus, it is important to assess whether the advice induced attrition in the sample. The third is death, one of the possible reasons for attrition. There were few deceased individuals after 4 years because the sample was still relatively young.

A second set of variables, presented in panel B in Table 2, corresponds directly to HBP indicators. Though there are often concerns about self-reported chronic diseases, self-report of hypertension diagnosis is generally accurate in developed countries [36, 37]. By definition, no individual at $t = 0$ reported having been diagnosed with HBP by a doctor or reported taking medication to lower BP. Very few individuals were taking BP-lowering medication without reporting the diagnosis. ELSA asked about medication only if HBP was reported or if another cardiovascular condition was reported. In the next wave, the proportion became 6% below and 15% above the cutoff after 1–2 years and after around 4 years 13% and 28%. For medication, the increase also appeared to be greater on the right-hand side of the cutoff. Those patterns were anticipated since higher SBP is expected to result in a higher probability of HBP diagnosis; however, the question is whether there was a jump at $t = 0$, which will be explored in the results section.

Panel C shows self reported health outcomes and biomarkers. Self reported diagnosis of diabetes and heart conditions (stroke, myocardial infarction, heart failure, or angina), self-reported good health, BMI and BP. As expected, health measures are slightly worse above the cutoff than below it.

Another standard consideration with RDD studies is the no manipulation of the running variable. That was not a particular concern in the present study since the selected respondents were unaware of the BP problem before the nurse visit. The reason is that they had not been diagnosed with HBP, and they were not expecting any sort of benefit from obtaining a record below the cutoffs. The formal tests and additional details are presented in Appendix C in S1 Appendix.

## 3 Results

### 3.1 Main results

Table 3 presents the differences at the cutoff for the variables of interest, based on local linear regressions with optimal bandwidths. Column 1 shows the differences at the study baseline: a balance test similar to that displayed in columns 8 and 9 of Table 1. Columns 2 and 3 in Table 3 show the differences after one wave (about 2 years) and two waves (about 4 years), respectively. Each cell shows the point estimate of the jump, and corrected P value in brackets. Also displayed are the optimal mean squared error (MSE) bandwidths used and sample size they cover to the left/right of the cutoff. Unless specifically stated below, similar results are obtained with alternative specifications with other bandwidths and polynomial orders or estimates that include covariates, and are presented in Appendix B in S1 Appendix. Figs 2–4 present graphic versions of the RDD analysis, organised in columns similar to the arrangement in Table 3. In all those graphs, the horizontal axis shows the sBP measurement, where 0 is the relevant cutoff. In the figures, each dot is the average of the variable of interest at a specific point of sBP (1 mmHg). Local quadratic smoothers appear as lines on either side of the threshold. The jump between those lines at $sBP = 0$ is the area of interest.

Panel A of Table 3 shows that there is neither differential attrition nor mortality below and above the threshold. Column 1 in Table 3 and the graphs in Fig 3, S1 and S2 Figs in S1 Appendix covering the same data indicate that there is no jump in any of the variables of interest before the nurse visit ($t = 0$). Likewise, there is no jump in columns 2 and 3 in Table 3 owing to the advice and subsequent visit to the doctor.

**3.1.1 HBP diagnosis and treatment.** Column 2 of panel B in Table 3 shows that the intervention did increase the likelihood of self-reported diagnosis of hypertension as well as the probability of being treated with medication for BP around 2 years after the nurse's advice. Among individuals aged ≤58 years at the time of the BP measurement, the following result was found: those above the sBP advice threshold were around 6.0 percentage points [p-val = 0.007] more likely to report they were taking medication and 5.7 percentage points [p-val = 0.060] more likely to report that a doctor diagnosed them with HBP. Both figures are large -especially the medication component: around 2% of the population with such BP levels takes medication below the cutoff, as shown in Table 2. Moreover, the findings are stable across specifications since modifying the bandwidths or polynomial order affects only standard errors.

It is important to recall that ambulatory measurements overestimate individuals' daily average sBP: that is one reason for the prevalence of hypertension not jumping to 100%. The other reason is that some patients did not follow the advice to visit their family doctor. Thus, the impact is the product of the following: (1) the proportion of the population at $sBP = 0$ that would normally not be diagnosed with HBP; (2) the true detectable HBP prevalence in such a

**Table 3. The effects of information on potential hypertension status.**

| Dependent Variable | (1) | (2) | (3) |
|---|---|---|---|
| | Before BP test | 2 years | 4 years |
| | $t = 0$ | $t = 1$ | $t = 2$ |
| **Panel A. Attrition** | | | |
| Missing this wave | | −0.005 | 0.018 |
| | | (0.029)[0.805] | (0.037)[0.540] |
| | | h = 14.8/12.7, N = 2041/938 | h = 13.0/13.5, N = 1737/979 |
| Already death by this wave | | −0.003 | 0.000 |
| | | (0.005)[0.485] | (0.008)[0.816] |
| | | h = 8.2/ 4.8, N = 977/395 | h = 14.2/ 8.9, N = 1866/644 |
| **Panel B. High Blood Pressure** | | | |
| Diagnosed HBP ever | | 0.057 | 0.040 |
| | | (0.027)[0.060] | (0.035)[0.384] |
| | | h = 15.8/14.0, N = 1874/846 | h = 18.8/15.2, N = 2031/787 |
| Takes BP medication | | 0.060 | 0.034 |
| | | (0.020)[0.007] | (0.025)[0.261] |
| | | h = 13.0/12.0, N = 1477/769 | h = 20.8/12.2, N = 2315/678 |
| **Panel C. Health** | | | |
| Diagnosed Diabetes ever | | 0.010 | 0.017 |
| | | (0.011)[0.375] | (0.019)[0.343] |
| | | h = 15.5/14.4, N = 1874/846 | h = 17.7/14.1, N = 1873/739 |
| Diagnosed Heart Condition | −0.002 | 0.018 | 0.036 |
| | (0.009)[0.787] | (0.017)[0.404] | (0.021)[0.161] |
| | h = 15.0/13.4, N = 2039/977 | h = 13.2/16.4, N = 1603/903 | h = 16.9/12.9, N = 1750/681 |
| Self-reported GOOD health | 0.016 | −0.007 | 0.022 |
| | (0.034)[0.624] | (0.035)[0.941] | (0.045)[0.630] |
| | h = 11.3/16.3, N = 1555/1094 | h = 12.1/18.4, N = 1465/958 | h = 9.5/15.1, N = 882/766 |
| Body Mass Index (BMI) | 0.120 | | 0.771 |
| | (0.412)[0.618] | | (0.548)[0.154] |
| | h = 13.9/14.7, N = 1810/980 | | h = 13.9/10.3, N = 1166/513 |
| Systolic BP (SBP) | −0.335 | | 0.026 |
| | (0.513)[0.518] | | (1.624)[0.815] |
| | h = 12.0/13.2, N = 1739/980 | | h = 15.8/16.3, N = 1232/606 |
| Diastolic BP (DBP) | −1.193 | | −0.336 |
| | (0.652)[0.115] | | (1.060)[0.765] |
| | h = 13.8/15.8, N = 1894/1065 | | h = 12.9/16.3, N = 966/606 |

Notes: This table present the impact estimates under several three specifications of a regression discontinuity design over the systolic blood pressure of respondents centred around 140 mmHg for ELSA. For males aged 50 and older in the HSE, the standardisation is done around 160 mmHg due to a different measurement protocol. It includes only those respondents aged 58 or younger at the time of the measurement who were not diagnosed with HBP or diabetes, and not taking BP-lowering medication. $h$ presents the optimal bandwidth to the left/right of the cutoff, and $N$ the corresponding number of observations effectively included.

Standard errors, in parenthesis, are derived from heteroskedasticity-robust nearest neighbour variance estimator with at least 3 neighbours. In the first set of brackets the conventional p-value is presented, while the second corresponds to the robust inference version derived by Calonico, Cattaneo, Titiunik (2014).

population—given that respondents with such an sBP level contact a family doctor, the likelihood of HBP being diagnosed—; and (3) the compliance rate with the advice. The first number can be extrapolated from the information just below the cutoff -around 95%. Accordingly, if all respondents who visited the doctor were diagnosed with hypertension, 6.25% of them

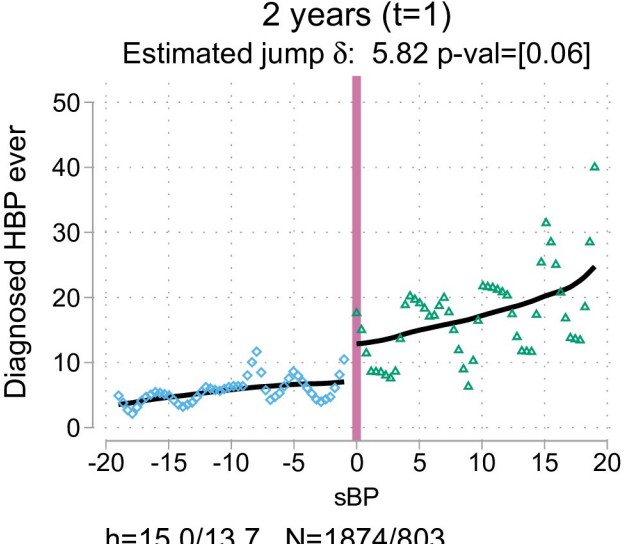

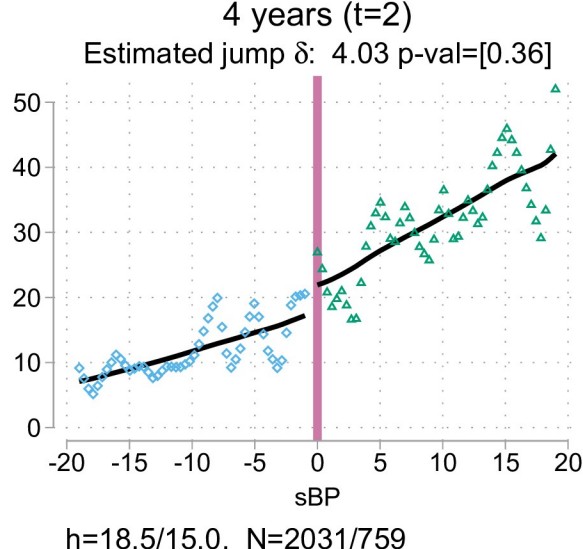

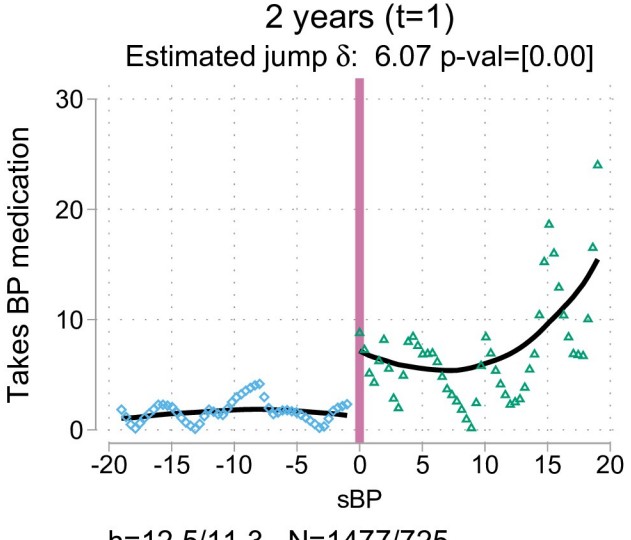

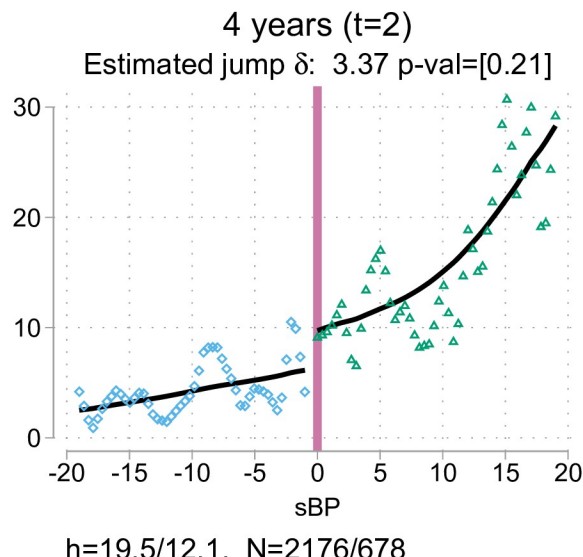

**Fig 2. Impacts on diagnosis of HBP and BP medication use.**

would have followed the advice. But if only 1 in 4 who visit a family doctor with this sBP level were diagnosed with HBP, then roughly 1 in 4 actually followed the advice of the nurses.

Column 3 of panel B in Table 3 presents the situation at 4 years: the magnitudes of the differences observed above were reduced; it cannot be rejected that they are equal to zero. As seen in Fig 2, the prevalence was almost 20% and medication use around 10% close to the cut-off; thus, the relevance of the differences also decreased. This result suggests that any gains from the advice would be based on the benefits derived from early detection of the condition. In other words, given the guidelines and primary care services, the advantage would derive

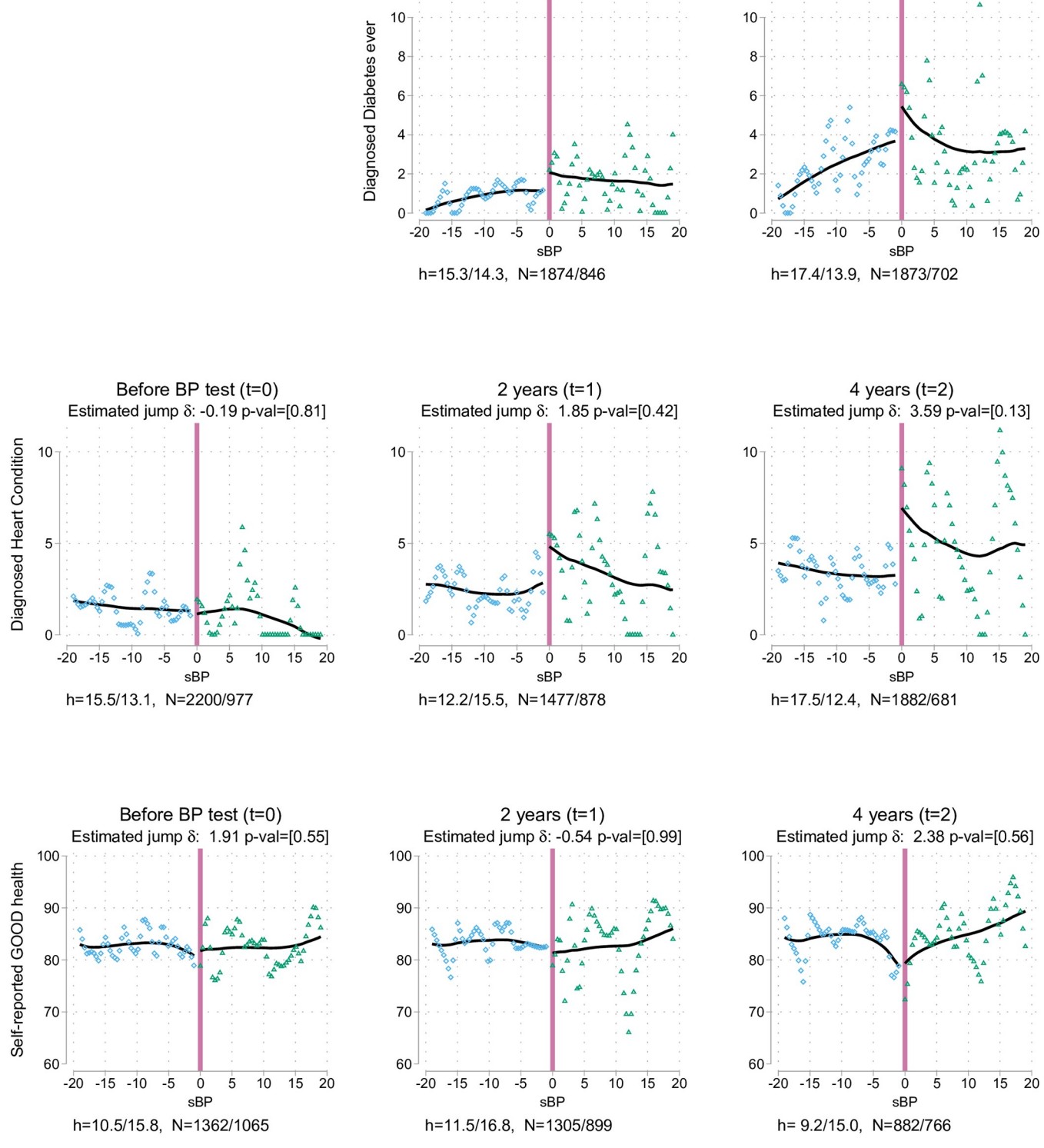

**Fig 3. Impacts on diagnosis of diabetes, heart conditions, and self-reported good health.**

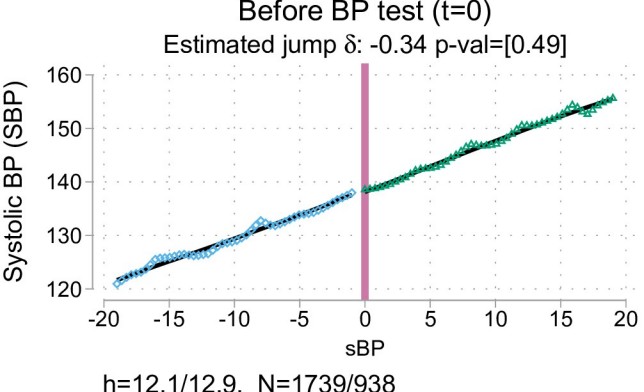

h=12.1/12.9,  N=1739/938

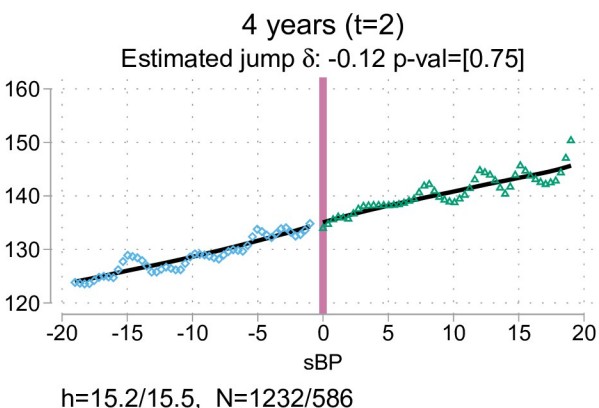

h=15.2/15.5,  N=1232/586

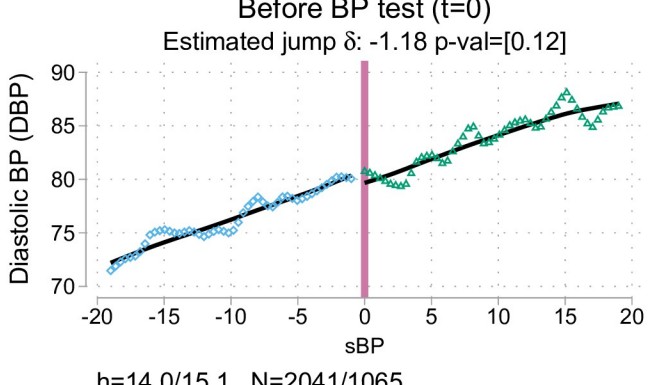

h=14.0/15.1,  N=2041/1065

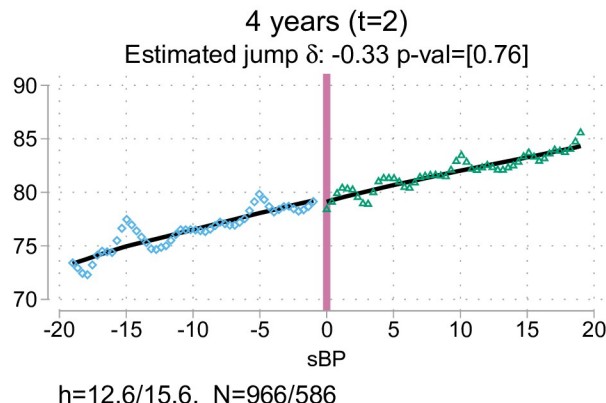

h=12.6/15.6,  N=966/586

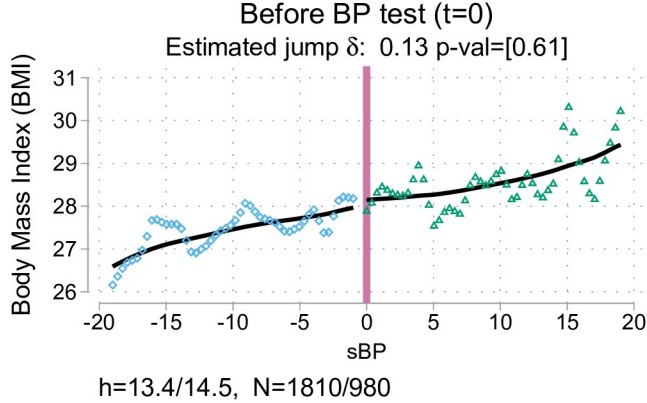

h=13.4/14.5,  N=1810/980

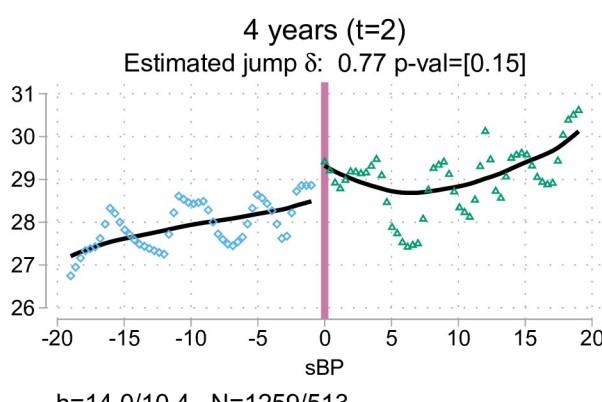

h=14.0/10.4,  N=1259/513

**Fig 4. Impacts on biomarkers.**

from detecting the condition a couple of years before the standard rate prevailing in the population.

**3.1.2 Health outcomes.**   The desired impact of any early diagnosis program is improving the health of the general population. In this case, the impact can be assessed by measuring whether diabetes, another common and generally asymptomatic condition, is detected. No differences were found in this respect: this finding indicates that at least at this level of BP, either under-diagnosed rates of diabetes type 2 were not particularly high or that family doctors did not find merit in testing for it. Further, no differences were evident in the onset of CVDs: that could be due to a mixture of low power for the prevalence of such conditions at the studied age and CVD risk and the rapid catch-up on the self-reported rate of hypertension for individuals below the threshold. With respect to perceived health, there was no evidence of an impact on reporting good health.

Finally, it is possible to assess if the advice produces a significant difference in BP and BMI. Given that biomarkers are collected every two waves, it is not possible to determine the short-term impact of the advice. Fig 4 shows the almost linear relationship between SBP and DBP at 4 years with the SBP at the initial measurement. Before the measurement, SBP was around 140 mmHg (by construction); 4 years later, it dropped to below 135 mmHg. However, there is no difference below and above the advice cutoff. For the case of BMI, the variable is still positively correlated with BP at the baseline measurement, showing its high persistence. There is no discontinuity in this relationship.

## 3.2 Heterogeneous effects

Heterogeneous effects by pre-intervention CVD-risk (using the Framingham risk score, a standard calculator used in GP practices) and self reported health were explored [38]. The impact on medication usage was higher for those with higher CVD-risk. Detailed results are presented in Appendix D in S1 Appendix. Finally, potential impacts at other points of the blood pressure distribution are explored. The results of this analysis are presented in Appendix D.1 in S1 Appendix, but unfortunately the sample size at this cut-off prevent a more detailed investigation.

## 3.3 Limitations

The data and methods used in this study result in several limitations that need to be considered for drawing conclusions.

First, results are specific for individuals in their late 50s and without previous diagnosis of cardiovascular diseases. Moreover, only for individuals with mildly elevated BP. However, this population is of particular interest in terms of policy analysis since those people are the ones most likely to be affected by health checks for that condition.

Second, the RDD method only uses a fraction of the information available in the study. Thus, it is possible that the non-rejection of the hypotheses is the result of lack of power of tests due to insufficient sample sizes. Appendix E in S1 Appendix presents power calculations of the main estimates following [39]. Current sample sizes allow us to detect impacts of half a standard deviation of BMI (test size of 0.956) and BP (test size of 0.99), four years after the BP measurements, and a bit lest for a quarter of standard deviations (test sizes around 0.45 to 0.69). However, for the diagnosis of conditions and the self-report of good health, it is hard to detect small changes (around 1 pp.) as the size of the tests are around 0.06–0.11 for most of them.

Third, the analysis here presented is restricted only to the final variables of BMI and blood pressure. Unfortunately, with the existing data, it was not possible to determine the

compliance rate with the advice (visit the GP), or the specific health behaviours that are modified. While ELSA has information on health behaviours, these are self reports and the specific questions change through time.

## 4 Conclusions and discussion

This study analysed the impact of health checks (which do not involve selection on demand for preventive care services) that advise individuals with BP around a certain threshold to follow up with a visit to their family doctor.

The first main question raised through this paper is the impact of advice on early detection of hypertension (hypothesis 1). The results showed a large, significant impact of the advice on the probability of receiving BP medication after having been diagnosed with HBP. However, the effect was temporal: after around 4 years, the difference cannot be distinguished from zero. While the test power might be limited for detecting small changes on diagnosis, the finding is confirmed by the continuity of BP levels at the cutoff in the nurse's follow-up visit. Essentially, the advice provided an early diagnosis (and treatment) of at least 2 years for around six of 100 respondents who were in the range of suspected hypertension stage 1. It also gave a warning to other respondents who did not obtain the diagnosis (or medication) but who became aware of being close to the limit.

The second aspect open for discussion is the impact on lifestyle (hypothesis 2). Guidelines suggest a lifestyle intervention that curbs smoking, poor dietary habits, and heavy alcohol consumption. The results show no impact on BMI, which suggests that any particular adjustment on lifestyle was either short-live or not enough to modify at least this risk factor. Results here are in line with findings for the *marginally ill* patients of diabetes in Japan and the US [12, 13]. No evidence of an impact on health behaviours or risk indicators, not only of advise but of a diagnosis, was found in those cases. For the case of Korea, only strong information shocks resulted in a change of behaviour and BMI [14].

Finally, patients who received the advice and those who do not have similar probabilities of being diagnosed with diabetes or with any heart condition (hypothesis 3). While the test has limited power to detect small variations on these outcomes, the results of hypotheses 1 and 2 suggest that hypothesis 3 would not hold.

A potential reason for the findings presented above is described by Kaestner and Lakdawalla when analysing the use of statins [40]. Individuals consider both medication and lifestyle adjustment as health investments, which could be either substitutes or complements. In our setting, given that blood pressure diagnosis is paired with medication use, respondents might not have a strong incentive to change their behaviours due to substitution between investments. As the case of mandatory health checks in Japan, in this study evidence of an increase of medication use with null long-run impacts on objective health indicators might indicate that *mild* information programs for the general population should be revised.

## Supporting information

**S1 Appendix.**
(PDF)

## Acknowledgments

I wish to express my gratitude for valuable comments from several anonymous referees, Dr. Jennifer Mindell, Eric French, Aureo de Paula, Suphanit Piyapromdee, Imran Rasul, Magne Mogstad, Luigi Siciliani, Uta Shöenberg, Marcos Vera-Hernandez and Victor Troster. I thank

the Edanz Group (www.edanzediting.com/ac) for editing a draft of this manuscript, and María Sofía Casabianca for her research assistance. Replication material can be found at https://github.com/androdri1/earlydiagnosis.

## Author Contributions

**Conceptualization:** Paul Rodríguez-Lesmes.

**Data curation:** Paul Rodríguez-Lesmes.

**Formal analysis:** Paul Rodríguez-Lesmes.

**Funding acquisition:** Paul Rodríguez-Lesmes.

**Investigation:** Paul Rodríguez-Lesmes.

**Methodology:** Paul Rodríguez-Lesmes.

**Project administration:** Paul Rodríguez-Lesmes.

**Software:** Paul Rodríguez-Lesmes.

**Validation:** Paul Rodríguez-Lesmes.

**Visualization:** Paul Rodríguez-Lesmes.

**Writing – original draft:** Paul Rodríguez-Lesmes.

**Writing – review & editing:** Paul Rodríguez-Lesmes.

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
