## [Decision Letter · Decision Letter 0]

25 Mar 2021

PONE-D-20-26199

Estimating the gains of early detection of hypertension over the marginal patient

PLOS ONE

Dear Dr. Rodriguez-Lesmes,

Thank you for submitting your manuscript to PLOS ONE. After careful consideration, we feel that it has merit but does not fully meet PLOS ONE’s publication criteria as it currently stands. Therefore, we invite you to submit a revised version of the manuscript that addresses the points raised during the review process.

We look forward to receiving your revised manuscript.

Kind regards,

Zhanjun Jia

Academic Editor

PLOS ONE

Journal Requirements:

Additional Editor Comments:

Please address the concerns of the experts.

Reviewers' comments:

Reviewer's Responses to Questions

**Comments to the Author**

1. Is the manuscript technically sound, and do the data support the conclusions?

Reviewer #1: Partly

Reviewer #2: Partly

2. Has the statistical analysis been performed appropriately and rigorously? 

Reviewer #1: I Don't Know

Reviewer #2: No

3. Have the authors made all data underlying the findings in their manuscript fully available?

Reviewer #1: Yes

Reviewer #2: Yes

4. Is the manuscript presented in an intelligible fashion and written in standard English?

Reviewer #1: Yes

Reviewer #2: No

5. Review Comments to the Author

Reviewer #1: General description: The manuscript by Paul Andres Rodriguez-Lesmes study the impact of early diagnosis on blood pressure, cholesterol, and body-mass-index by using the English Longitudinal Study of aging, in individuals not previously diagnosed with high Blood Pressure or diabetes nor are under pharmacological treatment. The Study use a regression discontinuity design encouraging patients that show above standard clinical BP threshold to visit their family doctor for a confirmatory diagnosis of hypertension. The data suggest a large, significant impact of the advice on the probability of receiving BP medication after having been diagnosed with high blood pressure. The advantage received from advising patients disappear over time (4 year), in which there were no difference in the parameters between the studied groups.

Major comments

-The manuscript does not state a clear hypothesis. The abstract does not state the hypothesis, and in the background, the hypothesis seems to be stated as “This paper studies the causal link between lifestyle and the acquisition of health information”. This is misleading, because to my understanding, the manuscript study whether early advise on health conditions (like high blood pressure, cholesterol and body mass index) has an impact on patient behavior (whether patients engage in taking care of themselves). In alignment with this comment, the conclusion should also be clear on whether the procedure has either positive, negative or no effect on patients/outcomes.

-The abstract seems to be unfinished, in which there is no conclusion and perspective of the findings.

-The manuscript does not have a clear and concise conclusion.

-Methods is an extension of the background.

-Advising people to visit the doctor and get medication seems to have a short-term improvement in preventing the rise of blood pressure. This may be due the fact that the behavior that lead to the rise in blood pressure is not address by doctors and that medication is not enough to prevent the expected outcome of such behaviors. So, the question is how the information provided in this manuscript contribute to the already known facts? This is important since the study has limitation regarding the lifestyle of the patients (diet, activity/sport, stress, etc).

-The author uses ELSA database containing data obtained during 2002-to-14 year. Does the database has more actual data? It is possible to include data from recent years?

-Grammar needs attention (for instance “The authors found that well-being is not affected the potential illness notification, but it is when respondents are classified to the in a risky position”).

Minor comments

-The manuscript is written in first person (I did, I found) while in general scientific-medical literature is written in technical and 3rd person.

-The author wrote “In this country, between 1994 and 2011, around 30% of adults aged 30 years and older had HBP”. This is confusing since the author is from Universidad Del Rosario Bogota, COLOMBIA. The author needs to clarify his statements and use more specific terms, like “In England, between….”.

Reviewer #2: 1)The manuscript is based on an undefined number of "N"s per test point. It is also not clear whether the succeeding subjects were the same as the original set of participants. For example, the incidence of the number of mortalities is not stated. Hence, the statistical formulas used may be sound but the tested population may not be appropriately sized.

2) The sample population was not indicated as British until later on in the manuscript. The source of the data used in this study may also not be technically freely available based on the fact that no mention of signed consent form is indicated. The only stipulation is the note from the author that some restrictions will apply since the source is from ELSA of NatCen.

3)The age group of the subjects studied is narrowly restricted to the 50 year old. The general findings are therefore restricted in this age group and cannot be taken as a general rule.

4) The follow-up of the parameters for the responses to the questions especially regarding the doctor visit upon "diagnosis" of HBP should be more strictly recorded and documented. There are too many mentions of other parameters such as findings in diabetic patients which can skew the results.

5) The writing of the manuscript needs to be reviewed and exclusively evaluated by a native English (U.S.) speaker.

6) First person pronouns such as "I" are never used or at least very strongly discouraged in research article writing.

6. PLOS authors have the option to publish the peer review history of their article (what does this mean?). If published, this will include your full peer review and any attached files.

Reviewer #1: No

Reviewer #2: No

---

## [Author Response · Author response to Decision Letter 0]

7 Jun 2021

Cover Letter

PONE-D-20-26199

Estimating the gains of early detection of hypertension over the marginal patient

Dear Dr Zhanjun Jia,

Dear Reviewers, 

I am pleased for the overall favorable reception of the manuscript and the opportunity to revise it. The detailed comments and suggestions by the two reviewers have been extremely helpful and incorporating them has greatly improved the accessibility of the paper. Below, I respond in detail to each comment and attach both a “clean” version of the revised manuscript as well as one that includes the “track changes”. 

Although the user agreement does not permit sharing the data directly, I include a link to a depository that includes detailed instructions on how the data can be requested with NatCen, as well as detailed replication materials (https://github.com/androdri1/earlydiagnosis). 

I hope that the substantial changes made to the manuscript are in line with the feedback from the reviewers and remain fully open to any further suggestions. 

Sincerely, 

The Author

Reviewer #1: 

Major comments

-The manuscript does not state a clear hypothesis. The abstract does not state the hypothesis, and in the background, the hypothesis seems to be stated as “This paper studies the causal link between lifestyle and the acquisition of health information”. This is misleading, because to my understanding, the manuscript study whether early advise on health conditions (like high blood pressure, cholesterol and body mass index) has an impact on patient behavior (whether patients engage in taking care of themselves). In alignment with this comment, the conclusion should also be clear on whether the procedure has either positive, negative or no effect on patients/outcomes.

Hypotheses of the study are implicit through the text and in the new abstract. Following this comment, now they are stated formally, and the conclusions refer to them directly.

H1: The advice would result in an increase in the diagnosis of HBP and on the treatment for it (ex. medication usage).

H2: The advice would result in a better lifestyle, reflected on lower levels of BMI (showing a change either on caloric intake or on exercise levels) and blood pressure.

H3: The advice would reduce the probability of the onset of other cardiovascular diseases, such as diabetes.

-The abstract seems to be unfinished, in which there is no conclusion and perspective of the findings.

Many thanks for pointing this out. The abstract was revised so it is in line with the journal standard in terms of content and length. Hopefully these elements are now clearer.

-The manuscript does not have a clear and concise conclusion.

The conclusion section is now shorter and refers directly to the hypotheses presented in the introduction. Other elements such as the limitations were incorporated in a separate section.

-Methods is an extension of the background.

Indeed, the name of material and methods section obeys to the style of the journal which states that this middle section should include everything that is not the introduction or the conclusion. As this paper does need a comprehensive explanation of how diagnosis of hypertension is carried on in the UK, and how the survey’s specific design changes such patterns, there is a large background component.

-Advising people to visit the doctor and get medication seems to have a short-term improvement in preventing the rise of blood pressure. This may be due the fact that the behavior that lead to the rise in blood pressure is not address by doctors and that medication is not enough to prevent the expected outcome of such behaviors. So, the question is how the information provided in this manuscript contribute to the already known facts? This is important since the study has limitation regarding the lifestyle of the patients (diet, activity/sport, stress, etc).

-The author uses ELSA database containing data obtained during 2002-to-14 year. Does the database has more actual data? It is possible to include data from recent years?

In the time between the preparation for submission, editor handling, and peer-review process, the last two waves of ELSA were published in the NatCen repository (waves 8 and 9, which cover data until 2019). I have updated the results of the document including the last available information. Yet, there are some minor elements to consider:

1. For waves 7,8 and 9, there is no mortality information. Therefore, it is not possible to expand the analysis for this outcome

2. Waves 8 and 9 have biomarkers, but they do not measure the height of respondents anymore. Therefore, in order to extend the analysis of BMI, height has been imputed from previous weights.

Aside from the points discussed above, there are no qualitative changes on the results.

-Grammar needs attention (for instance “The authors found that well-being is not affected the potential illness notification, but it is when respondents are classified to the in a risky position”).

Many thanks for pointing this out. The document has been reviewed carefully.

Minor comments

-The manuscript is written in first person (I did, I found) while in general scientific-medical literature is written in technical and 3rd person.

There were some sentences written in this fashion as I am an economist, and writing in the first person is not uncommon in the economics literature for specific methodological decisions or while explaining contributions to the literature. Yet, given that this paper is more likely to draw the attention of researchers in public health, all these phrases were rewritten.

-The author wrote “In this country, between 1994 and 2011, around 30% of adults aged 30 years and older had HBP”. This is confusing since the author is from Universidad Del Rosario Bogota, COLOMBIA. The author needs to clarify his statements and use more specific terms, like “In England, between….”.

The section of this paragraph is called “High blood pressure in the UK”. In addition, the previous phrase of the paragraph mentions the country: “(…) Such is the importance for preventive care, that some countries, such as the United Kingdom, provide financial incentives to primary care physicians for improving BP management”. However, to avoid any confusion, the phrase now says “In the UK, between (….).

Reviewer #2: 

1)The manuscript is based on an undefined number of "N"s per test point. It is also not clear whether the succeeding subjects were the same as the original set of participants. For example, the incidence of the number of mortalities is not stated. Hence, the statistical formulas used may be sound but the tested population may not be appropriately sized.

The design of the study is based on the longitudinal structure of the dataset. Therefore, the impact on outcomes of the individuals is based only on their own systolic blood pressure levels two or four years ago (see figure 1). Therefore participants are always the same through time. Yet, the referee has an important point with mortality as the selection of the survivors could shape the results. Panel A of Table 2 presents the mortality rates and the dropout rate of the survey without the number of observations as stated by the referee (this was added now to the table). In order to see if selection bias the estimates panel A of Table 3 considers the “impact” of the GP advice on attrition for both causes, including the specific sample size used for each regression. No evidence of selective attrition or mortality was found.

Another concern expressed here is the undefined “N”, which responds only to the requirements of the statistical method (optimal bandwidth selection of the regression discontinuity design). This implies that for each outcome, we are not comparing exactly the same individuals. Table 5 in the appendix considers a fixed bandwidth, so almost all regressions include the same number of individuals. As some variables are not available in all waves, there are some disparities. Results hold in general.

As for the “power” of the design, this is a valid concern. Following Cattaneo, Titiunik and Vasquez-Bare (2019), a power calculation test for the RDD design was implemented. Appendix E presents the power against impacts of 1 pp, 2.5 pp, and 5 pp for the binary variables; and ¼ standard deviation (SD), ½ SD, and 1 SD for the continuous variables. Detecting differential attrition is difficult (a test size of 0.316 for 5 pp, four years after the measurement), but the test seems adequate for mortality (0.68 for 2.5 pp, four years after the BP measurement). Concerning the main outcomes, it appears that with the given sample size, it is hard to detect impacts on HBP variables, showing the strength of the findings. As for health variables, the test has the power to detect changes of 5 to 10 mmHg on blood pressure and around 1.25 units of the BMI; while for self-reported good health and diagnosis of other conditions might be less suited. These limitations have been incorporated into the results and discussion section, which now is a separate sub-section of the document. Cattaneo MD, Titiunik R, Vazquez-Bare G. Power calculations for regression-discontinuity designs. The Stata Journal. 2019 Mar;19(1):210-45.

2) The sample population was not indicated as British until later on in the manuscript. The source of the data used in this study may also not be technically freely available based on the fact that no mention of signed consent form is indicated. The only stipulation is the note from the author that some restrictions will apply since the source is from ELSA of NatCen.

Many thanks for pointing this out. Now the first line of the abstract, and the first line of the second paragraph of the introduction, mention that study considers the case of England.

With respect to the data, the English Longitudinal Study of Ageing is a project developed by researchers of several UK academic institutions for at least two decades. It follows a standard pattern used by several studies in the world. Information is freely available upon registration with NatCen. I do not explain the survey's consent procedures and other characteristics as they are available on the webpage https://www.elsa-project.ac.uk/. A link to this website was added to the text.

3) The age group of the subjects studied is narrowly restricted to the 50 year old. The general findings are therefore restricted in this age group and cannot be taken as a general rule.

This is indeed a limitation of the study. Now is stated clearly on the new limitations sub-section.

4) The follow-up of the parameters for the responses to the questions especially regarding the doctor visit upon "diagnosis" of HBP should be more strictly recorded and documented. There are too many mentions of other parameters such as findings in diabetic patients which can skew the results.

I hope that it is now clearer that diagnosis of diabetes is used as an outcome for the third hypothesis. The data is restricted to individuals not diagnosed with such condition at the moment of the initial blood pressure measurement. As for the visit to the family doctor upon “diagnosis”, this is not captured by the survey.

5) The writing of the manuscript needs to be reviewed and exclusively evaluated by a native English (U.S.) speaker.

The Edanz Group proofread the article before submission, and they use native U.K. English speakers. I am deeply sorry that still there are mistakes on it. The current re-submitted version was reviewed carefully.

Apart from this, I could not find any PLOSONE’s style restriction that indicates that articles must be written in U.S. English. Therefore, it is still in British English.

6) First person pronouns such as "I" are never used or at least very strongly discouraged in research article writing.

There were some sentences written in this fashion as I am an economist, and writing in the first person is not uncommon in the economics literature for specific methodological decisions or while explaining contributions to the literature. Yet, given that this paper is more likely to draw the attention of researchers in public health, all these phrases were rewritten.

---

## [Decision Letter · Decision Letter 1]

24 Jun 2021

Estimating the gains of early detection of hypertension over the marginal patient

PONE-D-20-26199R1

Dear Dr. Rodriguez-Lesmes,

We’re pleased to inform you that your manuscript has been judged scientifically suitable for publication and will be formally accepted for publication once it meets all outstanding technical requirements.

Kind regards,

Zhanjun Jia

Academic Editor

PLOS ONE

Additional Editor Comments (optional):

Reviewers' comments:

Reviewer's Responses to Questions

**Comments to the Author**

1. If the authors have adequately addressed your comments raised in a previous round of review and you feel that this manuscript is now acceptable for publication, you may indicate that here to bypass the “Comments to the Author” section, enter your conflict of interest statement in the “Confidential to Editor” section, and submit your "Accept" recommendation.

Reviewer #2: All comments have been addressed

2. Is the manuscript technically sound, and do the data support the conclusions?

Reviewer #2: Yes

3. Has the statistical analysis been performed appropriately and rigorously? 

Reviewer #2: Yes

4. Have the authors made all data underlying the findings in their manuscript fully available?

Reviewer #2: Yes

5. Is the manuscript presented in an intelligible fashion and written in standard English?

Reviewer #2: Yes

6. Review Comments to the Author

Reviewer #2: The revision is a vast improvement to reply to the original criticisms brought about by the initial reviewers. The limitation of the available data source being analyzed is that the study was only performed on subjects from the UK. I would suggest that the title reflect to that fact and add UK in the title. Inferences to other population groups to show similar results may not hold true since different societies have different life standards and traditions. For example, Japanese adherence to directives from recognized authority is more seriously taken than other population sets. This may present a different impact on the outcome of a similar study. It was also mentioned that in the UK, small incentives were also provided to people to have a regular medical check-up. However, even with these incentives, the statistical results showed no significance after 4 years as compared to the control groups. This could add an additional variable to the mix.

It is also common fact that the use of RDD alone to predict the outcome of an assigned intervention makes it impossible to make true causal inference in the results as it does not reject the presence of potentially confounding variables.

There are some small typographical errors that have been found (lines 55, 68) so another round of perusal might be in order.

All in all, this is a great paper that is worthy to be published. It may make a big difference the health and economic well being of society.

7. PLOS authors have the option to publish the peer review history of their article (what does this mean?). If published, this will include your full peer review and any attached files.

Reviewer #2: No

---

## [Editor Report · Acceptance letter]

30 Jun 2021

PONE-D-20-26199R1 

Estimating the gains of early detection of hypertension over the marginal patient 

Dear Dr. Rodríguez-Lesmes:

I'm pleased to inform you that your manuscript has been deemed suitable for publication in PLOS ONE. Congratulations! Your manuscript is now with our production department. 

Kind regards, 

on behalf of

Dr. Zhanjun Jia 

Academic Editor

PLOS ONE